# Measurement of breast artery calcification using an artificial intelligence detection model and its association with major adverse cardiovascular events

**Suzanne J. Rose**[1], **Josette Hartnett**[2☯], **Zachary J. Estep**[3☯], **Daniyal Ameen**[4], **Shweta Karki**[1], **Edward Schuster**[5], **Rebecca B. Newman**[6]*, **David H. Hsi**[7]*

1 Department of Research and Discovery, Stamford Hospital, Stamford, Connecticut, United States of America, 2 Burke Rehabilitation, Montefiore Health System, 3 Sidney Kimmel Medical College at Thomas Jefferson University/Deborah Heart and Lung Center, Browns Mills, New Jersey, United States of America, 4 Department of Internal Medicine, Yale New Haven Health/Bridgeport Hospital, Bridgeport, Connecticut, United States of America, 5 Stamford Health Medical Group, Stamford, Connecticut, United States of America, 6 Department of Internal Medicine, Stamford Hospital, Stamford, Connecticut, United States of America, 7 Heart and Vascular Institute, Stamford Hospital, Stamford, Connecticut, United States of America

☯ These authors contributed equally to this work.
* rnewman@stamhealth.org (RBN); dhsi@stamhealth.org (DHH)

**Data Availability Statement:** We have made the data publicly available: https://doi.org/10.6084/m9.

## Abstract

Breast artery calcification (BAC) obtained from standard mammographic images is currently under evaluation to stratify risk of major adverse cardiovascular events in women. Measuring BAC using artificial intelligence (AI) technology, we aimed to determine the relationship between BAC and coronary artery calcification (CAC) severity with Major Adverse Cardiac Events (MACE). This retrospective study included women who underwent chest computed tomography (CT) within one year of mammography. T-test assessed the associations between MACE and variables of interest (BAC versus MACE, CAC versus MACE). Risk differences were calculated to capture the difference in observed risk and reference groups. Chi-square tests and/or Fisher's exact tests were performed to assess age and ASCVD risk with MACE and to assess BAC and CAC association with atherosclerotic cardiovascular disease (ASCVD) risk as a secondary outcome. A logistic regression model was conducted to measure the odds ratio between explanatory variables (BAC and CAC) and the outcome variables (MACE). Out of the 99 patients included in the analysis, 49 patients (49.49%) were BAC positive, with 37 patients (37.37%) CAC positive, and 26 patients (26.26%) had MACE. One unit increase in BAC score resulted in a 6% increased odds of having a moderate to high ASCVD risk >7.5% (p = 0.01) and 2% increased odds of having MACE (p = 0.005). The odds of having a moderate-high ASCVD risk score in BAC positive patients was higher (OR = 4.27, 95% CI 1.58–11.56) than CAC positive (OR = 4.05, 95% CI 1.36–12.06) patients. In this study population, the presence of BAC is associated with MACE and useful in corroborating ASCVD risk. Our results provide evidence to support the potential utilization of AI generated BAC measurements from standard of care mammograms in addition to the widely

figshare.26852719.v1. A statement to this nature has also been added to the manuscript under data collection.

**Funding:** The author(s) received no specific funding for this work.

**Competing interests:** The authors have declared that no competing interests exist.

adopted ASCVD and CAC scores, to identify and risk-stratify women who are at increased risk of CVD and may benefit from targeted prevention measures.

## Author summary

Breast arterial calcification, commonly seen during routine mammography, has been shown to be associated with adverse cardiac disease outcomes. Using artificial intelligence to estimate breast artery calcification, this study indicates the presence of breast artery calcification is significantly associated with Major Adverse Cardiac Events and may be beneficial in determining atherosclerotic cardiovascular disease risk. Artificial intelligence can be used to measure breast artery calcification in women undergoing routine mammography to identify women at increased risk of cardiovascular disease in need of targeted therapies and strategies.

## Introduction

Coronary artery disease (CAD) remains the leading cause of mortality in the United States (US), posing a significant health burden [1,2]. While CAD-related mortality has declined in men, it has not experienced substantial changes in women [3,4]. As seen in previous studies, with respect to the annual rate of first myocardial infarction (MI) by gender, men exhibit a significantly higher rate across all age groups compared to women [5]. However, a recent study on more than 45,000 patients hospitalized for a first heart attack found females were 20% more likely than males to develop heart failure or die within five years after their first heart attack, and presented at older ages with a greater burden of comorbidities [6]. This discrepancy highlights the need to address the under-utilization of early detection and treatment of CAD among women [7].

Traditional diagnostic tests that primarily focus on identifying CAD prove to be less effective in women than in men [8]. Consequently, there is a growing body of indirect evidence suggesting that prognostic risk assessment may serve as a more effective strategy than obstructive CAD assessment for women [9]. Similarly, more studies confirmed these findings, with the American Hospital Association (AHA) updating their guidance on assessing females for CVD risk factors [7].

Recently, a known association has been reported between breast artery calcification (BAC) and coronary artery calcification (CAC) [10,11], the latter of which has been shown to have an association with a 10-year atherosclerotic cardiovascular disease (ASCVD) risk [9]. Factors associated with a higher prevalence of BAC, apart from age, include diabetes mellitus, parity, chronic kidney disease, and personal history of CAD or risk equivalents [12,13]. CAC serves as a valuable prognostic tool for initiating lipid lowering therapy. Therefore, understanding the potential role of incidentally noted arterial calcification on mammography, recognizing the prognostic value, and appreciating the potential impact of reporting BAC for women's cardiovascular health are essential areas of investigation [10,14].

To aid healthcare practitioners in assessing cardiovascular risks in women, an investigational (at publication, FDA cleared for BAC detection and localization [15]), artificial intelligence (AI)-based software (cmAngio) was developed to identify BAC on mammograms, providing a proprietary BAC score, the investigational Bradley Score [16] from routine mammography [17]. This technology, if widely adopted, may create the potential for a novel heart

health screening program for patients undergoing annual mammography screening tests. When the cmAngio analysis is added to mammogram interpretation, breast cancer screening and vascular risk assessment can be performed without an additional visit, examination, or radiation to the patient [16]. The results could potentially help in early CVD risk assessment. The present study aims to determine the association between breast and coronary artery calcification and Major Adverse Cardiac Events (MACE).

## Study design and methods

### Study population

We performed a retrospective review of women aged 40–90 years old who were screened with full-field digital mammograms (FFDM) and were also referred for a standard non-contrast chest computed tomography (CT) within one year of receiving a mammogram at Stamford Hospital, CT, between January of 2013 and July of 2015. Final medical record and obituary review was conducted in January of 2024. Uncensored observations were included within the 11 years of follow-up as the study protocol was designed as retrospective and not established for time-to-event analyses. Exclusion criteria included patients with a known history of prior coronary revascularization, acute coronary syndrome, heart failure and valve replacement, patients with inadequate image quality due to significant respiratory motion artifacts, and patients with missing follow-up data. Patients with breast implants or less than 4-view mammograms were excluded.

### Ethics statement

Prior to study initiation, approval was received by Stamford Hospital's Institutional Review Board (IRB) of record (WCG IRB Work Order #1-1508938-1). Due to the retrospective nature of the research and anonymity of study data, the study was determined exempt from review by the IRB of record and therefore, consent was not required.

### Description of the Artificial Intelligence (AI) Model

BAC was quantified using a validated, proprietary investigational software (cmAngio, Cure-Metrix) built on a deep neural AI network. This model was trained on over 20,000 2D full-field digital mammograms (FFDM) from 13 healthcare facilities across Australia, Brazil, and the US, using an 80:20 train:tune data split, and a 60:40 BAC absent:BAC present split. The cmAngio software processes screening mammography images through the deep learning model to identify regions within the breast with a high probability of containing BAC. These regions are further analyzed, combining local and global imaging features such as density, contrast, and other physical dimensions to assess the presence and severity of BAC. Each of the four standard images is then assigned a score between 0 and 100, indicating the severity of BAC, with 0 representing no BAC and 100 representing the highest level of BAC.

To balance the algorithm's false positive and false negative rates, all image-level scores below 5 are floored to 0. The overall patient score, known as the BAC Bradley Score, is calculated as the mean of the threshold-adjusted image-level scores across all four views. BAC presence is determined if a patient has at least one image with BAC Bradley Score of 5 or higher. A binary case level output is then produced indicating BAC presence or absence. A previous study reported study results based on this threshold [18,19].

During development, each case was reviewed by two out of eleven Mammography Quality Standards Act (MQSA)-certified radiologists. The software demonstrated strong performance in detecting BAC, with an AUC of 0.98, a sensitivity of 94%, and a specificity of 96%. The

software has been approved by the FDA for BAC detection and is currently deployed in clinical settings, including large tertiary hospitals [15].

## Data collection

Patient demographic information, including age, ethnicity, all-cause mortality, and body mass index (BMI), was collected for all patients meeting inclusion criteria from the hospital's electronic medical records. CAC was defined as negative when patients were defined as low risk via Weston score, as previously described [20]. Additional clinical data, such as prior statin and aspirin prescription were also captured. The presence of comorbidities, such as hypertension, diabetes mellitus, dyslipidemia, smoking status, and prior CVD were obtained from medical history documentation. Calcium score collection methods has been previously reported [20]. ASCVD risk scores were calculated using the American Heart Association ASCVD risk calculator tool. Eleven patients had incalculable risk due to data outside of the allowable parameters. Patients were considered low risk if their calculated estimated 10-year ASCVD risk was under 5%, at moderate risk if scoring between 5–20%, and high risk if their results were over 20% [21]. All data has been made publicly available [22].

*Statistical Analysis*

All statistical analyses were performed in SAS version 8. Descriptive statistics summarizes the demographic and clinical characteristics of the study population. The prevalence of BAC was calculated as the number of patients with arterial calcifications divided by the total number of patients in the cohort. T-tests were performed to compare the mean Bradley and CAC scores among MACE (yes/no). Risk differences were calculated to capture the difference in observed risk of MACE with variables of interest (BAC, CAC and ASCVD risk). Odds Ratios (ORs) and their corresponding 95% confidence intervals (CI) were also calculated. ASVCD categories were collapsed to a binary variable; low risk (<7.5% risk) vs moderate—high risk (7.5%—over 20% risk). Chi-square tests were performed to assess age and ASCVD risk with MACE. Furthermore, a logistic regression model was conducted to measure the adjusted associations between explanatory variables (BAC and CAC) and the outcome variables (MACE) by significant confounders (age and ASCVD risk). The ORs obtained from the model for each explanatory variable was reported with the corresponding 95% confidence interval (CI) and *P*–value. Lastly, contingency tables with Chi-square tests and/or Fisher's exact tests were shown for the association between breast and coronary arterial calcifications with ASCVD risk score. Missing data were omitted, and available cases were included in the analysis. All analyses with resulting p-values <0.05 were considered statistically significant.

## Results

A total of 99 patients met inclusion criteria and were included in the analysis ranging from 47 years to 80 years (median age of 69). Table 1 shows the overall demographics and clinical characteristics. Out of the total study population, 47% were from the age group 70 and over years and 68.69% of them were white. Forty-nine (49) patients (49.49%) were BAC positive, and 37 patients (37.37%) were CAC positive. Fifty-nine patients (67.05%) had a moderate-high ASCVD risk score (7.5% to over 20%), and 26 patients (26.26%) had MACE. Out of the 99 patients, a total of 16 expired during the 11 year follow-up period.

Among patients who had MACE (n = 26), the mean BAC score was significantly higher (Mean BAC score = 35) compared to those without MACE (Mean BAC score = 15) (p = 0.01) (Table 2). Similar trend was found for CAC score for patients with MACE vs without MACE (p = 0.03) (Table 2).

**Table 1. Overall demographics and clinical characteristics (n = 99).** ASCVD = Atherosclerotic Cardiovascular Disease, BAC = Breast artery calcification, CAC = coronary artery calcification, MACE = Major Adverse Cardiac Events. *ASCVD risk score was incalculable for 11 patients due to data outside the allowable thresholds.

| Variable | Category | Count (%) |
|---|---|---|
| Age | *40–49* | 6 (6.06%) |
| | *50–59* | 16 (16.16%) |
| | *60–69* | 30 (30.30%) |
| | *70 and over* | 47 (47.47%) |
| Race | *White* | 68 (68.69%) |
| | *Black/ African American* | 16 (16.16%) |
| | *Hispanic/Latino* | 12 (12.12%) |
| | *Other* | 3 (3.03%) |
| BAC | *Positive* | 49 (49.49%) |
| | *Negative* | 50 (50.51%) |
| CAC | *Positive* | 37 (37.37%) |
| | *Negative* | 62 (62.63%) |
| ASCVD risk score * | *Low risk (<5%)* | 22 (25.0%) |
| | *Borderline risk (5%-7.4%)* | 7 (7.95%) |
| | *Intermediate risk (7.5%-19.9%)* | 30 (34.09%) |
| | *High risk (>20%)* | 29 (32.95%) |
| ASCVD risk score* (binary) | *Low risk (<7.5%)* | 29 (32.95%) |
| | *Moderate–High risk (7.5%—over 20%)* | 59 (67.05%) |
| MACE | *Yes* | 26 (26.26%) |
| | *No* | 73 (73.74%) |
| Death | *Yes* | 16 (16.16%) |
| | *No* | 83 (83.84%) |
| Smoking Status | *Never* | 57 (57.58%) |
| | *Current* | 38 (38.38%) |
| | *Unknown* | 4 (4.04%) |
| Diabetes | *Yes* | 20 (20.20%) |
| | *No* | 79 (79.80%) |
| Hypertension | *Yes* | 58 (58.58%) |
| | *No* | 41 (41.41%) |

The risk of MACE in BAC positive patients was 17% higher than BAC negative patients (95% CI 8%-33%). High risk ASCVD score patients had an 28% increased risk of MACE compared to those with low ASCVD risk score (95% CI 15% - 42%) (Table 3).

Out of the 26 patients who had MACE, 24 (92.31%) were over 60 years (p = 0.04) and 19 (95%) had moderate-high ASCVD risk score (p = 0.002) (Table 4). Age and ASCVD risk score were used as covariates in regression analysis in next table.

When assessed for the odds of MACE, the Bradley score (indicated as BAC) had 1.02 times (95% CI 1.01–1.04) and the CAC Score had 1.16 times the odds (95% CI 1.03–1.30). When

**Table 2. Comparison of mean BAC and CAC scores among MACE (yes/no).** BAC = Breast artery calcification, CAC = coronary artery calcification, MACE = Major Adverse Cardiac Events.

| Variable | MACE (Mean (SD)) | | T-test P-value |
|---|---|---|---|
| | *No (n = 73)* | *Yes (n = 26)* | |
| CAC Score | 2.05 (3.12) | 4.19 (4.55) | 0.03 |
| BAC Score | 15.14 (25.82) | 35.27 (35.29) | 0.01 |

**Table 3. Risk value and risk difference of MACE on BAC, CAC and ASCVD risk score (n = 99).**
ASCVD = Atherosclerotic Cardiovascular Disease, BAC = Breast artery calcification, CAC = coronary artery calcification, CI = Confidence Interval.

| Variable | Category | Risk value (95% CI) |
|---|---|---|
| CAC | *No* | 0.21 (0.09, 0.33) |
| | *Yes* | 0.30 (0.18,0.42) |
| **CAC Absolute Risk Difference** | | 0.09 (-0.08, 0.26) |
| BAC | *No* | 0.18 (0.7–0.28) |
| | *Yes* | 0.35 (0.21–0.48) |
| **BAC Absolute Risk Difference** | | 0.17 (0.08–0.33) |
| ASCVD risk score | Low risk (<7.5%) | 0.03 (0.0,0.10) |
| | Moderate–High risk (7.5%—over 20%) | 0.32 (0.20,0.44) |
| **ASCVD risk score Absolute Risk Difference** | | 0.28 (0.15,0.42) |

adjusted for age and ASCVD risk score, Bradley score (indicated as BAC) had a statistically significant effect on MACE with OR = 1.02 (1.0–1.04) however, the odds of CAC score were not significant (OR = 1.14 (95% CI = 0.99–1.32) (Table 5).

Among those who had a moderate-high ASCVD risk score (n = 59), significant differences were found in CAC and BAC results. Forty-six percent of moderate-high ASCVD risk score patients were CAC positive as compared to 54% CAC negative (p = 0.009) (S1 Table). Similarly, 58% of ASCVD moderate-high risk scoring patients had BAC positive compared to 42% BAC negative (p = 0.003) (S1 Table). Assessing the odds of having a moderate-high ASCVD risk score, BAC positive patients had higher odds (OR = 4.27, 95% CI 1.58–11.56) than CAC positive (OR = 4.05, 95% CI 1.36–12.06) (S2 Table). With each unit increase in Bradley score (shown as BAC), there was 6% increased odds of having a moderate to high ASCVD risk score (p = 0.01) (S2 Table). The mean BAC score among those who had moderate-high ASCVD risk score was significantly higher (27.4) than those with low-risk score (3.8) (p = <0.0001) (S3 Table).

## Discussion

To the best of our knowledge, our study is the first to report on the utilization of artificial intelligence to demonstrate a positive correlation between a positive BAC score and MACE. The presence of BAC in this population was significantly associated with MACE. The BAC

**Table 4. Chi-square test showing association of age and ASCVD risk score with MACE.** Atherosclerotic Cardiovascular Disease, MACE = Major Adverse Cardiac Events.

| Variable | Category | MACE | | Chi Square P-value |
|---|---|---|---|---|
| | | *No (n = 73)* | *Yes (n = 26)* | |
| Age | <60 years | 20 (27.40%) | 2 (7.69%) | *0.04* |
| | 60 years and over | 53 (72.60%) | 24 (92.31%) | |
| ASCVD risk score | Low risk (<7.5%) | 28 (41.18%) | 1 (5%) | 0.002 |
| | Moderate–High risk (7.5%—over 20%) | 40 (58.82%) | 19 (95%) | |

**Table 5. Crude and multivariable logistic regression models for the association of BAC and CAC and covariates with MACE (n = 99).** ASCVD = Atherosclerotic Cardiovascular Disease, BAC = Breast artery calcification, CAC = coronary artery calcification, CI = Confidence Interval, MACE = Major Adverse Cardiac Events.

| | Univariate (crude) Models | | | Multivariate Model adjusted by age and ASCVD risk score | | |
|---|---|---|---|---|---|---|
| | Odds Ratio | 95% CI | P-value | Odds Ratio | 95% CI | P-value |
| BAC | 1.02 | 1.01–1.04 | 0.005 | 1.02 | 1.00–1.04 | 0.002 |
| CAC | 1.16 | 1.03–1.30 | 0.01 | 1.14 | 0.99–1.32 | 0.06 |

numbers are closely aligned with the ASCVD data in being significantly associated with MACE, which already has a well-known association with greater morbidity and mortality [23]. In addition, an increase in BAC score had a statistically significant correlation with the presence of MACE. In order to align the results with ASCVD risk score as a strong predictor of lifetime risk of cardiovascular events [21], ASCVD risk score was assessed as an outcome. As expected [24], CAC was found to be significantly associated with all factors in the ACVD data analysis, with BAC proving to be non-inferior as a predictive modality.

The association of BAC with CAC is shown in multiple studies [11] although there is a greater probability for a mammogram to be available for most women starting at the age of 40 versus a calcium score from conventional CT coronary angiography. Importantly, this study emphasizes the importance of considering BAC, as a surrogate marker for ASCVD in risk assessment for MACE in women patients. If BAC positivity correlates with an increased risk of MACE, early intervention such as lifestyle modifications and statin medications can be initiated early for primary prevention of cardiovascular events. It is important to recognize some unique pathophysiology for women including soft plaque rupture, vasoconstriction, microvascular angina, and ischemia with no obstructive coronary arteries [25]. Based on the large volume of data, it may be more appropriate to consider CT coronary angiography, but not CT calcium scores in women patients with elevated ASCVD risks in the future [25,26]. Our research did not focus on the complex social issues such as gender identity.

While primary care can offer important healthcare screenings for women, such as cancer, diabetes, heart disease and osteoporosis, national trends indicate that primary care visits have shown a 20% decrease in the past decade, while specialists rates have increased [27–30]. Although women are more likely to report seeing a primary care physician on an annual basis, they frequently do not see heart specialists [28]. In contrast, almost 40 million women undergo routine mammography screening in the US annually [31].

The utilization of BAC as a potential marker for CAD and the risk of MACE can have a potential impact on public health. Our results highlight that the positive BAC and moderate to high ASCVD risk score correlates strongly with MACE. A recent study reported that physicians, if given the ability to have more information on BAC, preferred to receive this type of reporting as it could impact patient care [32]. Indeed, traditional cardiac risk scoring and CAC scores may not be the best gauge of heart disease in women as studies have demonstrated that women classified as low-risk by Framingham Risk Score demonstrated significant CAC scores at high percentiles [10]. In the present study, the inclusion of BAC significantly improved the classifier's ability to predict CAD when considering risk factors, suggesting BAC contains valuable information for CAD prediction that is not captured by age alone. Our findings of CAC by itself not predicting MACE in women seem to be supported by PROMISE and SCOT-HEART data that in symptomatic women with non-obstructive CAD, the Agatston calcium score was not independently associated with MACE. These authors also found that age and the ASCVD risk score were both independently associated with future risk of MACE, but not CACs [26].

## Limitations

A key limitation is the use of non-gated CT chest to classify CAC and not a gated cardiac CT. However, our previous results, did attest to the utility of non-gated CT to measure coronary artery calcification using the Weston method developed by Cleveland Clinic [20]. We are in the current process of adopting a new AI based algorithm for CAC quantitation in the future. In addition, a relatively small sample size lends to the inability to fully generalize the results. Prospective and large studies need to be initiated to study women who undergo routine mammograms without known cardiac risk factors over an extended period to assess long-term outcomes. The accuracy of AI calculated BAC will need to be verified by more investigators in the future for accuracy and reproducibility. BAC at the current time remains an investigational tool, which can potentially augment CV risk prediction in women patients, in addition to the guideline supported CAC and ASCVD risk score.

## Conclusion

Utilizing AI-assisted BAC assessments may offer a novel and noninvasive approach to identify women at increased risk of cardiovascular disease who could benefit from targeted prevention strategies.

## Supporting information

**S1 Table. Association of BAC, CAC with ASCVD risk score (n = 99).**
ASCVD = Atherosclerotic Cardiovascular Disease, BAC = Breast artery calcification, CAC = coronary artery calcification. *Fisher's exact test.
(DOCX)

**S2 Table. Odds Ratios of moderate-high ASCVD risk score (n = 99).**
ASCVD = Atherosclerotic Cardiovascular Disease, BAC = Breast artery calcification, CAC = coronary artery calcification, CI = Confidence Interval, OR = Odds Ratio.
(DOCX)

**S3 Table. Mean Score among ASCVD risk score (yes/no) (n = 99).**
ASCVD = Atherosclerotic Cardiovascular Disease, BAC = Breast artery calcification, CAC = coronary artery calcification, SD = Standard Deviation.
(DOCX)

## Acknowledgments

We would like to acknowledge Richard Mantey, MS, Junhao Wang, MS, Homa Karimabadi, PhD, Kevin Harris and Danielle Naaman at CureMetrix for their steadfast support and encouragement in the writing of this manuscript.

## Author Contributions

**Conceptualization:** Suzanne J. Rose, Josette Hartnett, Zachary J. Estep, Daniyal Ameen, Shweta Karki, Edward Schuster.

**Data curation:** Suzanne J. Rose, Josette Hartnett, Zachary J. Estep, Daniyal Ameen.

**Formal analysis:** Suzanne J. Rose, Josette Hartnett.

**Investigation:** Suzanne J. Rose, Josette Hartnett, Zachary J. Estep, Daniyal Ameen, Rebecca B. Newman, David H. Hsi.

**Methodology:** Suzanne J. Rose, Josette Hartnett, Shweta Karki, Edward Schuster, Rebecca B. Newman, David H. Hsi.

**Project administration:** Suzanne J. Rose, Josette Hartnett, Rebecca B. Newman, David H. Hsi.

**Supervision:** Rebecca B. Newman, David H. Hsi.

**Validation:** Suzanne J. Rose, Josette Hartnett.

**Writing – original draft:** Suzanne J. Rose, Josette Hartnett, Zachary J. Estep, Daniyal Ameen.

**Writing – review & editing:** Suzanne J. Rose, Josette Hartnett, Shweta Karki, Edward Schuster, Rebecca B. Newman, David H. Hsi.

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
