## [Decision Letter · Decision Letter 0]

19 Aug 2024

PDIG-D-24-00204

Measurement of Breast Artery Calcification Using an Artificial Intelligence Detection Model and its Association with Major Adverse Cardiovascular Events

PLOS Digital Health

Dear Dr. Rose,

Thank you for submitting your manuscript to PLOS Digital Health. After careful consideration, we feel that it has merit but does not fully meet PLOS Digital Health's publication criteria as it currently stands. Therefore, we invite you to submit a revised version of the manuscript that addresses the points raised during the review process.

Please submit your revised manuscript within 60 days Oct 18 2024 11:59PM. If you will need more time than this to complete your revisions, please reply to this message or contact the journal office at digitalhealth@plos.org. Please include the following items when submitting your revised manuscript:

We look forward to receiving your revised manuscript.

Kind regards,

Dhiya Al-Jumeily OBE, PhD

Section Editor

PLOS Digital Health

Journal Requirements:

1. We do not publish any copyright or trademark symbols that usually accompany proprietary names, eg (R), (C), or TM (e.g. next to drug or reagent names). Please remove all instances of trademark/copyright symbols throughout the text, including ® and ™ on pages 17 and 7.

2. We noticed that you used "data not shown" in the manuscript. We do not allow these references, as the PLOS data access policy requires that all data be either published with the manuscript or made available in a publicly accessible database. Please amend the supplementary material to include the referenced data or remove the references.

3. In the online submission form, you indicated that your data will be submitted to a repository upon acceptance. We strongly recommend all authors deposit their data before acceptance, as the process can be lengthy and hold up publication timelines. Please note that, though access restrictions are acceptable now, your entire minimal dataset will need to be made freely accessible if your manuscript is accepted for publication. This policy applies to all data except where public deposition would breach compliance with the protocol approved by your research ethics board. If you are unable to adhere to our open data policy, please kindly revise your statement to explain your reasoning and we will seek the editor's input on an exemption.

Additional Editor Comments (if provided):

The review process has been lingering and I apologize for this delay. 

I have now received two contrasted reviews, Reviewer 1 suggesting to carry out a substantial rewritting.

To the reviewers' comments, I would also add the request to enhance the manuscript in order to highlight better the significance of the study, and also to address the various points below:

Was the study protocol and statistical analysis plan registered before the start of the study?

Having tables split over two pages make it difficult to read. Please make sure this does not happen in any resubmission.

cover page: I do understand the word "objective" in the sentence "Objective Breast artery calcification (BAC) obtained from standard mammographic

images is currently under evaluation to stratify risk of major cardiovascular events in

women."

Abstract p.2 spell out ASCVD and include this variable in the objctive section.

"Contingency tables, Chi-square tests and/or Fisher's exact

tests assessed the associations between arterial calcifications and categorical variables of interest" the result of those test are not reported nor commented in the main text.

Key messages p.4: "It has previously been reported an association exists between breast artery calcification and coronary artery calcification, the latter of which is associated with coronary artery disease". 

In what way does a message reported in earlier studies constitute a key message of the present study?

"Using artificial intelligence to measure breast artery calcification"

If artificial intelligence is used, I would call it an "estimate" rather than a "measurement".

p.6 Study Design and Methods

- What is the follow up period for the observation of MACE?

- How was censoring handled?

p.7 Description of the Artificial Intelligence (AI) Model

Is the ML approach explained in more detail elsewhere?

p. 8-9 "Contingency tables, chi-square tests and/or Fisher's exact tests were

performed to assess the association between breast and coronary arterial calcifications and categorical variables of interest (BAC vs MACE, CAC vs MACE, ASCVD risk vs MACE)."

There are four variables involved here, hence up to six pairwise comparisons. 

What was the rationale for carrying out only those three comparisons?

p.9 "Risk differences were calculated to capture the difference in observed risk between the reference groups and the groups with the exposure of interest."

At this stage of my perusal, it was not clear "the exposure of interest" referred to.

p.9 "univariate and multivariate logistic regression analysis was performed with MACE as an outcome to see the association of BAC and CAC adjusted by significant confounder"

Please explain in more detail what you have done here.

p. 10-11 It is not clear what Table 2 contains. 

I have the impression that the right most column contain risk values and risk difference values. Please enhance wording in cells and caption. 

Caption mentions "ASCVD" in the sentence "Risk difference of MACE on BAC, CAC and ASCVD" but I have the impression that throughout the paper, it is referred to "ASCVD risk".

p. 11-12 as pointed out earlier, the description of the logistic regression modelling lacks details

Reviewers' comments:

Reviewer #1: In this retrospective study, Rose and colleagues retrospectively studied 99 patients who underwent a non-contrast CT chest within one year of mammography at Stamford Hospital between January 2013 and July 2015 to study the association BAC obtained using AI (via cmAngio) and MACE as well as CAC and MACE. In their abstract/summary, they report that the odds of having a moderate-high ASCVD risk was 3.5 in the BAC positive group and 4.05 in the CAC positive group.

I recommend restructuring the study to compare the diagnostic utility of ASCVD risk score vs BAC vs CAC in MACE prediction. Associations between ASCVD risk and BAC/CAC are not of clinical utility in my opinion.

Also, a key limitation here is that non-gated CT chest was used to classify CAC, not a gated cardiac CT.

Reviewer #2: The authors of this manuscript reported a retrospective study on the association between major adverse cardiovascular events and breast artery calcification score obtained using an AI program from standard screening mammograms. BAC scores are found to be significantly associated with MACE, and are no-inferior to CAC scores in prediction of the ACVD risks. The manuscript is well written, and the topic is of interest to the readers of this journal. The reviewer recommends publication with minor revisions. Please find the detailed comments below: 

1. About the AI model in the method section: more details should be included about this model. For example, the authors claimed that over 20,000 mammograms were used in the training. What percentage of those images has BAC? How was the true BAC score determined for the training data set. Is there any validation at all for the test data set used in this study?

2. The authors did not clarify it, but it seems to suggest that both 2D and TOMO images were used for BAC assessment in this study. Were there any differences in the findings, in regard to the use of 2D or TOMO images? 

3. The authors used a Bradley Score of 5 to classify the BAC images into a binary category. But there was no discussion about how this threshold was chosen? Is the associations between BAC and MACE sensitive to this threshold setting?

4. The authors suggested that each unit increase in Bradley score in BAC has a 6% increase of odds in ASCVD. How about the increased risk of MACE?

5. In page 14, the authors claimed that the combination of positive BAC and positive ASCVD correlates strongly with MACE. However, the reviewer was not able to find direct results to support this conclusion.

---

## [Decision Letter · Decision Letter 1]

10 Nov 2024

Measurement of Breast Artery Calcification Using an Artificial Intelligence Detection Model and its Association with Major Adverse Cardiovascular Events

PDIG-D-24-00204R1

Dear Dr. Rose,

We are pleased to inform you that your manuscript 'Measurement of Breast Artery Calcification Using an Artificial Intelligence Detection Model and its Association with Major Adverse Cardiovascular Events' has been provisionally accepted for publication in PLOS Digital Health.

Best regards,

Gilles Guillot

Academic Editor

PLOS Digital Health

Reviewer #1: All comments have been addressed

Reviewer #2: All comments have been addressed